**Seasonal shifts in depth to water uptake by young thinned and overstocked lodgepole pine**
**(*Pinus contorta*) forests under drought conditions in the Okanagan Valley, British**
**Columbia, Canada**
Emory C. Ellis[1], Robert D. Guy[2], Xiaohua A. Wei[3]
[1]School of Forestry, Northern Arizona University, Flagstaff, Arizona, 86001, USA
[2]Department of Forestry and Conservation Sciences, University of British Columbia, Vancouver, British Columbia,
V6T1Z4, Canada
[3]Department of Earth, Environmental and Geographic Sciences, University of British Columbia (Okanagan
Campus), Kelowna, British Columbia, V1V 1V7, Canada
Correspondence to: Emory C. Ellis (ece58@nau.edu)
**Abstract:**
As drought and prolonged water stress become more prevalent in dry regions under climate
change, preserving water resources has become a focal point for maintaining forest health. Forest
regeneration after forest loss or disturbance can lead to over-stocked juvenile stands with high
water demands and low water-use efficiency. Forest thinning is a common practice with the goal
of improving tree health, carbon storage, and water use while decreasing stand demands in arid
and semi-arid regions. However, little is known about the impacts of stand density on seasonal
variation in depth to water uptake nor the magnitude of the effect of growing season drought
conditions on water availability. Existing reports are highly variable by climatic region, species,
and thinning intensity. In this study, stable isotope ratios of deuterium ($\delta^2$H) and oxygen ($\delta^{18}$O)
in water collected from soil varying depths and from branches of lodgepole pine (*Pinus contorta*)
under different degrees of thinning (control: 27,000 stems per ha; moderately thinned: 4,500
stems per ha; heavily thinned: 1,100 stems per ha) over the growing season were analyzed using
the MixSIAR Bayesian mixing model to calculate the relative contributions of different water
sources in the Okanagan Valley in the interior of British Columbia, Canada. We found that under
drought conditions the lodgepole pine trees shifted their depth to water uptake through the
growing season (June to October), to rely more heavily on older precipitation events that
percolated through the soil profile when shallow soil water became less accessible. Decreased
forest density subsequent to forest thinning did not cause a significant difference in isotopic
composition of branch water but did cause changes in the timing and relative proportion of water
utilized from different depths. Thinned lodgepole pines stands were able to maintain water
uptake from 35 cm below the soil profile whereas the overstocked stands relied on a larger
proportion of deep soil water and groundwater towards the end of the growing season. Our
results support other findings by indicating that although lodgepole pines are drought tolerant
and have dimorphic root systems, they did not shift back from deep water sources to shallow soil
water when soil water availability increased following precipitation events at the end of the
growing season.
Keywords: *Pinus contorta*; stable water isotopes; forest thinning; water-use strategies;
preferential water uptake; dual-isotope analysis; Bayesian isotope mixing model; soil water
uptake; transpiration; the interior of British Columbia

## 1. Introduction

As forests recover after harvesting, carbon and water demands change, and future climate projections of increased drought severity will further complicate biogeochemical cycling and carbon-water trade-offs (Giles-Hansen et al., 2021; Wang et al., 2019). Overpopulated regenerating stands can add further stress on ecosystems; for example, light competition in dense juvenile stands increases stand water demands by driving vertical growth and canopy cover (Liu et al., 2011). To mitigate this stress, management strategies such as systemic thinning of high-density juvenile stands have been shown to promote forest regeneration while decreasing competition and providing remaining vegetation with increased light availability, rooting space, nutrient access, and space for horizontal branch growth (Giuggiola et al., 2016). Over a variety of forest ecosystems, reductions in stand density increase light availability, tree water use, carbon storage, and water-use efficiency, an indication of improved tree health, and to decrease stand water use, reducing the intensity of water stress under drought conditions (Belmonte et al., 2022; Fernandes et al., 2016; Giuggiola et al., 2016; Liu et al., 2011; Manrique-Alba et al., 2020; Molina & del Campo, 2012; Park et al., 2018; Sohn et al., 2012, 2016; Wang et al., 2019). Because the primary goal of forest thinning is to decrease stand water use and increase productivity, papers reporting the effects of this management strategy often focus on changes in carbon storage, tree growth, transpiration, and water-use efficiency (Giuggiola et al., 2016; Manrique-Alba et al., 2020; Park et al., 2018; Sohn et al., 2016). However, few studies have reported sources of water use for vegetation water uptake and shifts in depth to water uptake in association with thinning treatments in overstocked naturally regenerating forests, particularly under drought conditions.

Quantifying stand water use is imperative to predicting the future of water availability in our ecosystems. However, various studies indicate that trees do not always use the most recent precipitation, and that vegetation can utilize different sources of water at different soil depths depending on availability or stress (Dawson & Pate, 1996; Grossiord et al., 2017; Wang et al., 2017). Many studies also report the depth of water uptake of various species and the relationship between co-existing species and shared water sources (Andrews et al., 2012; Brinkmann et al., 2019; Grossiord et al., 2017; Langs et al., 2020; Liu et al., 2015; Maier et al., 2019; Meinzer et al., 2007; Sánchez-Pérez et al., 2008; Szymczak et al., 2020; Wang et al., 2017; Warren et al., 2005). In water-limited regions such as arid and semi-arid landscapes, some species have adapted to derive water from various depths over time depending on seasonal water variability, indicating higher ecological plasticity and drought tolerance (Langs et al., 2020; Wang et al., 2017). Understanding where in the soil profile plants obtain water, over prolonged dry periods and at different stand densities, is essential in assessing the impact of forest thinning and the relative importance of different seasonal water sources during shifts in water availability in arid regions and under future climate conditions (Evaristo et al., 2015; Prieto et al., 2012; Sohn et al., 2016). The implications of depth to water uptake and seasonal changes in water utilization, in conjunction with water-use efficiency, can emphasize the importance of the timing and volume of precipitation events and primary contributors to vegetation water use.

Stable isotope ratios can be used as powerful natural tracers to identify distinct water sources such as rainfall, snow, and groundwater (Brinkmann et al., 2018; Lin & Sternberg, 1993; Sprenger et al., 2017; Stumpp et al., 2018). The isotopic signature of precipitation events is altered by elevation, temperature, and evaporative fractionation creating distinctive layers within

the soil profile (Kleine et al., 2020; Sprenger et al., 2017; Stumpp et al., 2018). More
specifically, soil water reflects precipitation events as they infiltrate through the soil layer with
the influence of evaporative fractionation until mixing with older soil water and groundwater and
creating individualized water isotopic signatures throughout the soil profile (Andrews & Science,
2009; Brinkmann et al., 2018; Dawson & Pate, 1996; Sprenger et al., 2017; Stumpp et al., 2018).
The isotopic composition of plant water can correspond to the water uptake depth in the soil
profile (Brinkmann et al., 2019; Langs et al., 2020; Meinzer et al., 2007; Stumpp et al., 2018;
Wang et al., 2017). Due to these unique characteristics, stable water isotopes have been used by
researchers to assess sources of water used by plants and their possible shifts under altered
environmental conditions (Evaristo et al., 2015; Flanagan & Ehleringer, 1991; Meinzer et al.,
2001; Stumpp et al., 2018).
Lodgepole pine (*Pinus contorta* Douglas) is an early successional montane conifer with a deep
tap root, fine roots in shallow soil layers, and an adventitious rooting system which allow this
species to access water throughout the soil profile (Fahey & Knight, 1986; Halter & Chanway,
1993). Depending on the species, root structures have two main components; namely, lateral
roots to exploit soil near the surface, and, in species with dimorphic root systems, sinker roots or
a well-developed tap root to reach deeper soil water or groundwater when surface water is
limited. Species with dimorphic rooting systems can access water from different depths in the
soil profile depending on soil moisture content and water availability, making them more
resilient to water scarcity or prolonged drought conditions (Dawson & Pate, 1996; Meinzer et al.,
2013). Wang et al. (2019) studied the short-term effects of thinning overstocked juvenile (16-
year-old) lodgepole pine stands in the Upper Penticton Creek Watershed, British Columbia,
Canada, and found a significant positive relationship between growth and water use from
decreased stand density and that heavily thinned treatments showed the most drought resistance.
Andrews et al. (2012) compared water uptake strategies between Douglas-fir (*Pseudotsuga*
*menziesii* (Mirb.) Franco) and lodgepole pine in southern Alberta, and found that lodgepole pines
are able to minimize seasonal variations in stem water potential and that tap roots are deep
enough to access groundwater. These finding are consistent with other literature reporting that
decreased stem density can improve water-use efficiency and that conifer trees can access water
from different depths depending on moisture availability (Meinzer et al., 2007a; Warren et al.,
2005). The literature therefore indicates that lodgepole pines can access water from different soil
layers even under extreme or prolonged drought conditions, but little is known about the shifting
of water use under different stand densities as a result of thinning treatments and drought
conditions.
In this study, we build on the research from Wang et al. (2019) which looked at the effects of
thinning on water-use efficiency during a drought and non-drought year by analyzing the stable
isotope ratios ($\delta^2$H and $\delta^{18}$O) of soil and xylem water to evaluate at what depths overstocked and
thinned stands access water over a growing season to further our understanding of the
ecosystem-level impacts of thinning as a management strategy. We hypothesized that lodgepole
pine primarily relies on spring snowmelt, but reductions in shallow source water during the
growing season would drive trees to utilize deeper sources of water as the season progressed. We
also hypothesized that decreased stand density (thinning) would increase shallow soil
evaporation due to decreased canopy cover, but also decrease competitive limitations in tree
rooting zones so that at lower densities trees could better maintain mid-level soil water uptake.
Through a detailed partitioning of tree water sources, we can better understand how lodgepole
pine uses water, estimate proportional dependence of lodgepole pine on specific source waters,
and determine if thinning affects tree water use and uptake strategies under drought conditions.

2. Methods
2.1. Study site
The study was conducted in the Upper Penticton Creek experimental watershed (UPC) northeast
of Penticton in the interior of British Columbia, Canada (49°39'34" N,119°24',34" W). The site
elevation is approximately 1675 m with steep, rocky terrain and a southern aspect (Wang et al.,
2019). The luvisolic soils were formed from granite; the texture is coarse sandy-loam and is well
drained with a low water holding capacity (Hope, 2011; Winkler et al., 2021; Winkler & Moore,
2006). The biogeoclimatic region is the Engelmann Spruce-Subalpine Fir zone with cold, snowy
conditions from November to early
June and seasonal drought
conditions during the summer
months, June to October (Coupe et
al., 1991; Wang et al., 2019). This
research site was initially
established as a paired watershed
experiment in the early 1980s to
quantify the impact of forest
harvesting on water resources
(Creed et al., 2014; Moore &
Wondzell, 2005; Winkler et al.,
154 2021).

The juvenile thinning experiment
began in 2016 when 16-year-old,
evenly aged, regenerating lodgepole
pine stands were thinned to different
densities than a control (Control - C:

*Figure 1 Watershed location and treatment plots of moderately thinned (T1), heavily thinned (T2), and the controlled (C) over-populated stands across the three replicate blocks (Wang et al., 2019)*

27,000 stem ha$^{-1}$, T1: 4,500 stems ha$^{-1}$, and T2: 1,100 stems ha$^{-1}$) where C represents the control
stands, T1 represents the moderately thinned stands, and T2 represents the heavily thinned stands
(Figure 1). The three treatments were repeated across three replicate blocks. Each block was 75
m long and 25 m in width with three 20 m$^2$ plots and 5 m between treatment plots. After the
initial thinning, all debris was left on site.
2.2. Climate and soil moisture monitoring
Climate stations (HOBO weather station, Onset Computer, Bourne MA, USA) were deployed
across Block 1 treatments and have measured meteorological data since 2016 (ambient
temperature, relative humidity (rH), wind speed, precipitation, and solar radiation) in 10-minute
intervals. From these data, we calculated daily vapor pressure deficit (VPD) as well as daily and
monthly potential evapotranspiration (PET) (Flint & Childs, 1991; Russell, 1960; Streck, 2003).
Recorded historical precipitation (1997-2008) was acquired from a long-term climate station in a
lodgepole pine forest in the 241 experimental watershed (climate station P7) (Moore et al.,
173 2021).

Rainfall and temperature data from Block 1 were related to historical data to calculate the
monthly dryness (PET/P), standardized precipitation index (SPI), and standardized precipitation
evapotranspiration index (SPEI) (Table S1) (Beguería et al., 2014; Stagge et al., 2014; Wu et al.,
2005). In the middle of the growing season in 2021, four soil moisture probes (HOBO TEROS
11 Soil Moisture/Temp Probes) were deployed in each treatment in Block 1 to measure changes
in soil moisture and temperature at 5 cm and 35 cm at 15-minute increments (n=12).
2.3 Sample collection
We sampled three trees per treatment across the three blocks and three in an adjacent mature plot
south of the study site four times over the 2021 growing season in approximately six-week
intervals (June 11-12, July 21-22, September 10-11, and October 7-8) around noon to capture
peak transpiration time (Table 1). We used a pole pruner to cut a mid-canopy branch in the live
crown. We peeled the bark off branch segments with no needle coverage to remove outer bark
and phloem, placed them into 10 mL glass tubes that were then with Parafilm® wrap, covered in
aluminum foil, and set in a cooler until the end of the day when they were transferred to a freezer
at -18˚C. During the last two sampling periods, some trees had red needles, likely an indication
of dryness or higher temperatures from an early growing season heat dome that began in June.
*Table 1 Overview of the branch, soil, and precipitation samples collected over the four sampling periods during the*
*2021 growing season and additional campaigns to collect groundwater and stream water.*

| Sample Type | | | Sampling Period | 1 | 2 | 3 | NA | 4 |
|---|---|---|---|---|---|---|---|---|
| | | | Sampling Date | June 11-12 | July 21-22 | September 10-11 | October 1 | October 9 |
| | | | Branches | 33 | 33 | 33 | 0 | 33 |
| | Soils | | *5* | 9 | 9 | 9 | 0 | 9 |
| | | | *20* | 0 | 6 | 0 | 0 | 0 |
| | | | *35* | 9 | 9 | 9 | 0 | 9 |
| | | | *40* | 0 | 6 | 0 | 0 | 0 |
| | | | *60* | 0 | 6 | 0 | 0 | 0 |
| | | | *80* | 0 | 6 | 0 | 0 | 0 |
| | | | *100* | 0 | 6 | 0 | 0 | 0 |
| | | | Rain | 1 | 0 | 1 | 0 | 0 |
| | Precipitation | Snow | | 1 | 0 | 0 | 0 | 1 |
| | | | Stream | 0 | 0 | 0 | 8 | 0 |
| | | | Groundwater | 0 | 0 | 0 | 6 | 0 |


Soil samples were collected horizontally from 40 cm soil pits randomly dug within each
treatment plot at 5 and 35 cm depths from the surface from June to October of 2021. Large rocks
were removed from the profile. We conducted soil ribbon field tests to ensure that clay
composition was less than 10% (soil ribbons were less than 20 mm in length). Soils were taken
directly from the pit, then sealed in freezer seal bags and frozen until cryogenic distillation for
water extraction. In July, 1 m pits were dug. From the vertical pit, samples were collected in 20
cm increments to determine the depth of tree water access. After samples were collected, the
larger rocks and soils were used to fill the pits. We assumed that the isotopic signature of soil
water below 40 cm would be similar throughout the growing season and would be representative
of deep soil water. Soil samples were stored in a freezer at -18˚C until cryogenically distilled.
Precipitation samples were collected cumulatively over individual field collection days where
precipitation was present (Table 1). Snow from a late spring event was collected on June 11[th] to
represent snow water isotopic composition during the sublimation and melt period of early 2021.
Another snow event was collected on October 11[th] during an active snowfall. A rain event was
collected on September 10[th]. Groundwater and stream samples were collected from the creek 241
watershed in early October 2021 at the beginning of the seasonal hydraulic recovery period
(Table 1). Groundwater was collected using a hand pump. Groundwater and stream samples
were collected at the end of the growing season as stream beds were dry and groundwater was
inaccessible during the dry period. Once the well had been pumped and cleared, 10 mL glass test
tubes were rinsed with ground water three times before being filled. Precipitation, groundwater
and stream samples were collected into 10 mL glass test tubes, sealed with Parafilm® and foil,
and stored in a fridge at 4ºC.
2.4 Cryogenic extraction and isotopic analysis
Before extraction, branch samples remained sealed and were weighed in the glass test tubes used
for field collection. Branches remained in the test tubes until cryogenic distillation was complete
to ensure that any liquid water lost from the branch to the test tube was contained in the extract.
Soils samples were mixed in the Ziploc® bag, weighed, and transferred to a glass round bottom
flask. For stable isotope analysis, water was extracted from stem and soil samples using
cryogenic distillation (Orlowski et al., 2013; Pearcy et al., 2012). The test tube and branch
sample segment of the line was immersed in liquid nitrogen for 10 minutes until frozen
(Chillakuru, 2009). Soil sample size for extraction was roughly determined based on the
expected moisture of the frozen sample and soil moisture readings from continuous
measurements in the field. Soils were frozen for 45 minutes in a 500 mL round-bottom flask
using a dry-ice and 95% ethanol mixture before pumping out the air. Frozen samples were
pumped down to 60 mTorr, not disturbing the sample (Tsuruta et al., 2019). The vacuum-sealed
extraction unit was detached from the pump and transferred to a boiling water bath; the
extraction tube was submerged in liquid nitrogen. Branch samples were set to distill for 1 hour
and soil samples for 2 hours or until the tubing was clear to ensure all mobile and bound source
water was extracted (Orlowski et al., 2013; Tsuruta et al., 2019; Vargas et al., 2017; West et al.,
2006). As reviewed by Allen & Kirchner (2022), the cryogenic vacuum distillation of water from
plant tissues and soils can cause systematic biases in the measurements of $\delta^2H$. The degree of
extraction bias varies depending on species and soil type (Allen & Kirchner, 2022).In contrast,
bias in $\delta^{18}O$ values is close to zero (Allen & Kirchner, 2022). Reported biases in $\delta^2H$ average
about -6.1‰ for xylem water and -4‰ for water extracted from sandy soils, such as the soils
sampled here, which are of similar magnitude (Allen & Kirchner, 2022). Therefore, although we
used cryogenic vacuum distillation to extract water from xylem and soil media, potential
systematic bias introduced during the extraction process was treated as negligible asall sources
we identified had a difference in $\delta^2H$ greater than 4‰ (with the minimum distance being 14‰
between groundwater samples and deep soil water), minimizing any major effects on partitioning
calculations.
The volume of branch water extracted ranged from 1 to 3 mL depending on the size of the
branch sample. Total extracted water varied dependent on the mass of the initial sample. The
volume of soil water extract ranged from 1mL to 7 mL depending on the size of the sample
prepared for extraction. Samples were also weighed after extraction and compared to oven dried
samples to ensure distillation was complete. Water extracted from branch and soil samples
accounted for 47.9±3.2% and 9±6% of mean sample weight ± standard deviation.
All samples were pipetted and sealed into glass vials with screw tops and shipped to the
University of California Davis Stable Isotope Facility (Davis, CA, USA) for $^{18}$O and $^{2}$H analysis
using headspace gas equilibration on a GasBench-II interfaced to a Delta Plus XL isotope-ratio
mass spectrometer (Thermo-Finnigan, Bremen, Germany) normalized to a range of secondary
reference waters calibrated against three IAEA standard waters.  Precision was less than or equal
to 2.0‰ for $\delta^{2}$H and 0.2‰ for $\delta^{18}$O. Results were returned in the "delta" notation expressing the
isotopic composition of each sample as a ratio in parts per thousand, relative to VSMOW
(Vienna-Standard Mean Ocean Water) where:
$$\delta(‰) = \left( \frac{R_{\text{Sample}}}{R_{\text{Standard}}} - 1 \right)$$

Sample extract was situated in an isotope biplot and compared to the global meteoric water line
(GMWL) along with a local meteoric water line for the Okanagan Valley (OMWL) ($\delta^{2}$H = 6.6
($\delta^{18}$O) - 22.7) and local evaporative line (LEL) ($\delta^{2}$H = 5 ($\delta^{18}$O) - 48.4) calculated for the
Okanagan Valley by Wassenaar et al. (2011). The LEL is a linear regression that indicates the
departure of water sources from the OMWL to indicate the degree of evaporative processes
fractionating the isotopic composition of water sources or variance in the isotopic composition of
seasonal precipitation events.
One extreme outlier of B1C at the 20 cm depth was removed before analysis; the high $\delta^{2}$H and
$\delta^{18}$O values were likely due to contamination or incomplete cryogenic distillation. To test the
variance between thinning treatments, block replicates, sampling periods, and soil depth, we first
tested the assumption of normality in the subsets using the Shapiro-Wilk test and found that all
subgroups were approximately normally distributed. Repeated measures ANOVAs were used to
compare effects of date and treatment on $\delta^{2}$H and $\delta^{18}$O in branches, soils and groundwater to
determine if changes in lodgepole pine uptake patterns occurred over time, if soil signatures
varied between different depths (0-100 cm and groundwater) and densities, and if thinning
juvenile stands changed seasonal shifts. All statistical analysis was conducted in R Studio
(version 1.3.1073) using the appropriate tests to determine site distinctions and seasonal
variability in depth to uptake (RStudio Team, 2020).
2.4 MixSIAR model scenarios
Process-based models (PBM) with a Bayesian approach include integrating other processes or
existing information as priors allowing for a more informed approach than a simple linear model
(Ogle et al., 2014). To accurately partition potential lodgepole pine water sources, we used the
MixSIAR modeling package, a Bayesian mixing model (BMM) based on the Markov Chain
Monte Carlo method (MCMC) (Langs et al., 2020; Stock, 2013/2022, p. 201; Stock et al., 2018;
Wang et al., 2017; Wang et al., 2019). The MixSIAR modeling package was selected over
previous iterations of the dual-isotope BMM (SIAR and Simmr) and other partitioning models
because of the accuracy in the analysis of covariates and the ability of the model to include
source-specific uncertainties and discrimination factors (Stock et al., 2018; Wang et al., 2017).
We partitioned potential water sources for five different scenarios using a combination of single
and dual isotope approaches and different potential sources: scenario 1 – single isotope $\delta^{18}O$ two
sources 5 cm and 35 cm depth; scenario 2 – single-isotope $\delta^2H$ two sources 5 cm and 35 cm
depth; scenario 3 – dual-isotope two sources 5 cm and 35 cm depth; scenario 4 – dual isotope
three sources 5 cm, 35 cm and 45-100 cm depth; scenario 5 – dual isotope three sources 5 cm,
35-100 cm and groundwater; and scenario 6 – dual isotope four sources 5 cm, 35 cm, 45-100 cm
and groundwater. In scenarios using deep
soil water (35-100 cm depths), the isotopic
composition was calculated as a weighted
average between seasonally collected soil
water from depth 35 and average soil
water at depths collected in 202 cm
intervals during the early growing season
(n=38 per season). There were no source
concentration dependencies, and the
discrimination was set to zero for both
isotopes in the analysis. The run length of
the Markov chain Monte Carlo (MCMC)
was set to 'normal' (chain length =
100,000; burn =50,000; thin = 50; chains
= 3). The Gelman-Rubin and Geweke
diagnostic tests included in the model
package were used to determine
convergence (Gelman-Rubin score <
1.01). Scenarios that did not converge
were run again with a longer runtime (chain
length: 300,000; burn: 200,000; thin: 100;
chains = 3). No priors were used, so each
water source was considered equally (α =
315 1).

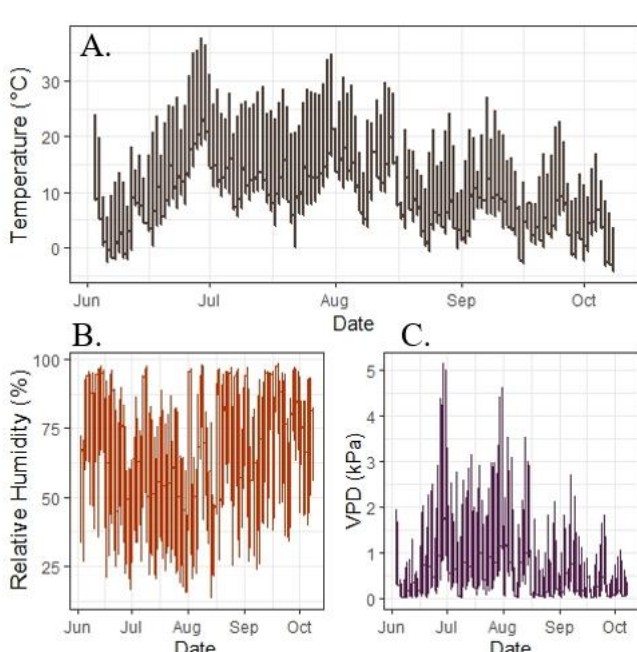

*Figure 3 15-minute measurements of A. atmospheric temperature (˚C), B. Relative humidity (%), and C. vapor pressure deficit (VDP) (kPa).*


3. Results
3.1. Climate and soil water content

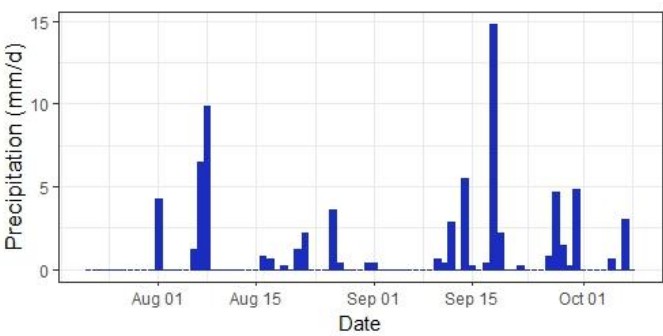

*Figure 2 Rainfall (mm/d) from July 22 to October 8, 2021.*

The ambient temperature peaked in
the moderately thinned plot (T1)
on June 29[th] with a maximum
temperature of 36.3°C in an
abnormally hot and dry summer
(Figure 2). Relative humidity and
VPD recorded in T1 showed the
most variability and highest
evaporative capacity during July.
Atmospheric water vapor was
higher in late September and
October when precipitation was
more frequent, and the watershed
began to exhibit traits of
hydrologic recovery (Figure 3).
One indication of increased water
availability was increased soil
moisture at 5 cm and 35 cm depths
and more groundwater recharge in
October (Figure 4). There was
17.5 mm of precipitation from
September 16[th] to 18[th] that
infiltrated to at least 35 cm below
the soil surface along with

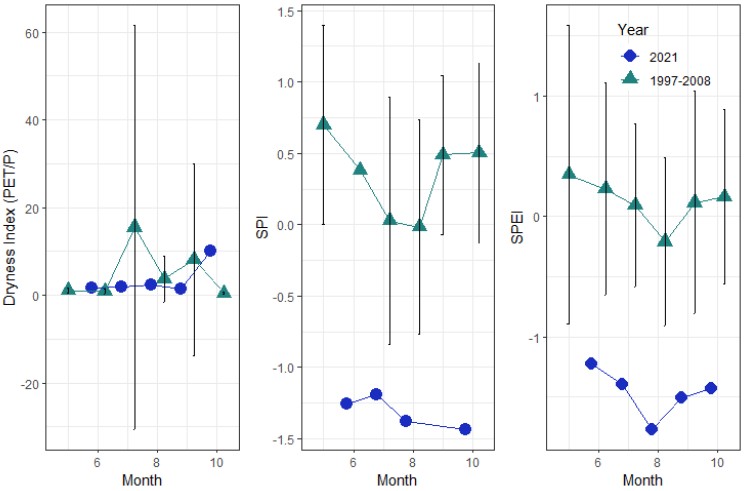

*Figure 5 From left to right: dryness index (monthly PET using the Thornthwaite method divided by mean monthly precipitation) from June to October, 2021 and historic climate data from 1997 to 2008 including the including standard error for the historic climate data, standard precipitation index (SPI) with a 3-month period from June to October of 2021 and historic (1997-2008) climate data including standard error for the historic climate data, and standardized precipitation evapotranspiration index (SPEI) with a 3-month period from June to October of 2021 and historic (1997-2008) climate data including standard error for the historic climate data.*

subsequent rainfall events that likely infiltrated past the 35 cm sample depth changing the
isotopic composition of deep soil water from what was measured during the deep pit sampling in
July.
Rainfall events recorded at a nearby long-term research station between June to October from
1997-2008 represented approximately 30.1% of annual precipitation (Winkler et al., 2021). Over
the 2021 study period, there was 147.8 mm of rainfall, while the mean summer rainfall from
1997 to 2008 was 232.5 mm, and most of the rainfall occurred in the early growing season. SPI
and SPEI were significantly lower in 2021 than the mean historical range (Figure 5). Although
there was precipitation and the beginning of hydraulic recovery in October, drought conditions
persisted. Drought conditions of the study site reflected the drought conditions of the region as
reported by Agriculture and Agri-Food Canada from June to August 2021 in moving from severe
(level 2 drought) to exceptional (level 4) before recovering in September (Canada, 2014:
https://agriculture.canada.ca/en/agricultural-zproduction/weather/canadian-drought-
monitor/drought-
analysis).
3.2. Water stable isotopes

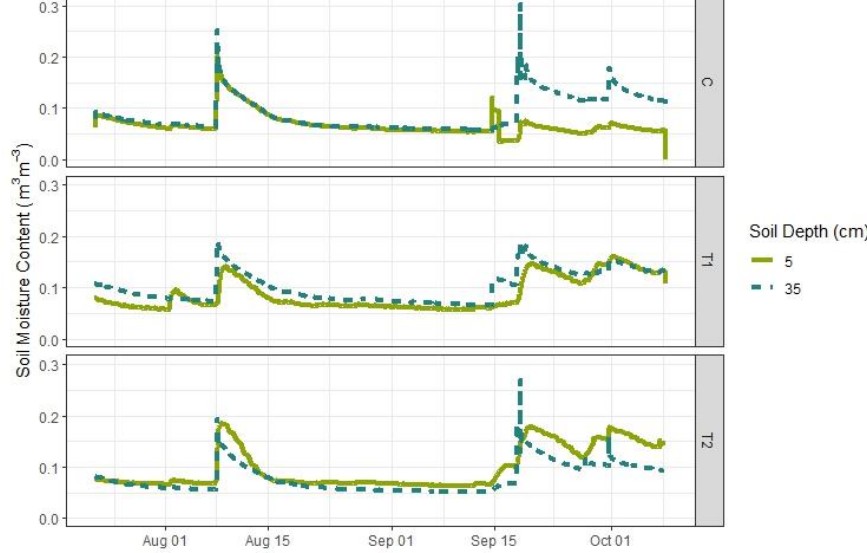

The biplot of sample
isotopic composition
shows the distribution and
effect of isotopic
fractionation on source
water isotope ratios of
samples collected during
the 2021 field season.
Field collected samples
were compared to the
Okanagan Meteoric
Water Line (OMWL)
(Wassenaar et al., 2011).
The slopes for branch and
soil water were less steep
than the OMWL, and the

*Figure 5 Average in-situ continuous measurements (15-minute interval) of soil water content ($m^3/m^3$) from the control, moderately thinned, and heavily thinned stands in Block 1.*

intercepts more negative, indicating that evaporative fractionation contributed to the isotopic
composition of these pools at the UPC (Figure 6). Soil samples seemed to follow the LEL
produced by Wassenaar et al. (2011) for the region indicating similar evaporative fractionation
effects. Branch water more closely following the OMWL than soils, suggesting that most
samples consisted of water that was accessed from deeper in the soil profile and had infiltrated
past the evaporative front. Precipitation samples collected during the field season fell along the
OMWL (Wassenaar et al., 2011). The $\delta^2H$ and $\delta^{18}O$ of the June 11[th] rainfall event were -127.5‰
and -13.03‰, respectively. The September rainfall event was much more enriched with a $\delta^2H$ of
-38.4‰ and $\delta^{18}O$ of -2.89 (Figure 6). The snowfall collected on October 7[th] more closely
resembled the lighter, colder, June precipitation event.
3.2.1. Soil moisture and seasonal water composition
Soil moisture probes and percent soil water content from samples collected for isotopic analysis
were compared between treatments and deployment depths. Water content of soil samples was
highest in June (21.5% at 5 cm and 21.6% at 35 cm) because of high snow melt and early spring
precipitation, while soils were driest in September (6.32% at 5 cm and 6.19% at 35 cm).
Continuous soil moisture measurements showed that soil water began to increase in mid-
September as precipitation became more frequent, daily solar radiation decreased, and water
percolated into deeper soil layers. There were significant differences in the continuously
measured soil moisture by depths, treatments, and month, respectively (5-35 cm) (Depth: F
=3545.9, p<2e-16***) (Treatment: F=1883.3, p<2e-16***) (Month: F=3359.8, p<2e-16***)
(Figure 7), but soil water content of samples for isotopic analysis only varied significantly by
month (August – October) (F=22, p<5.4e-9***).

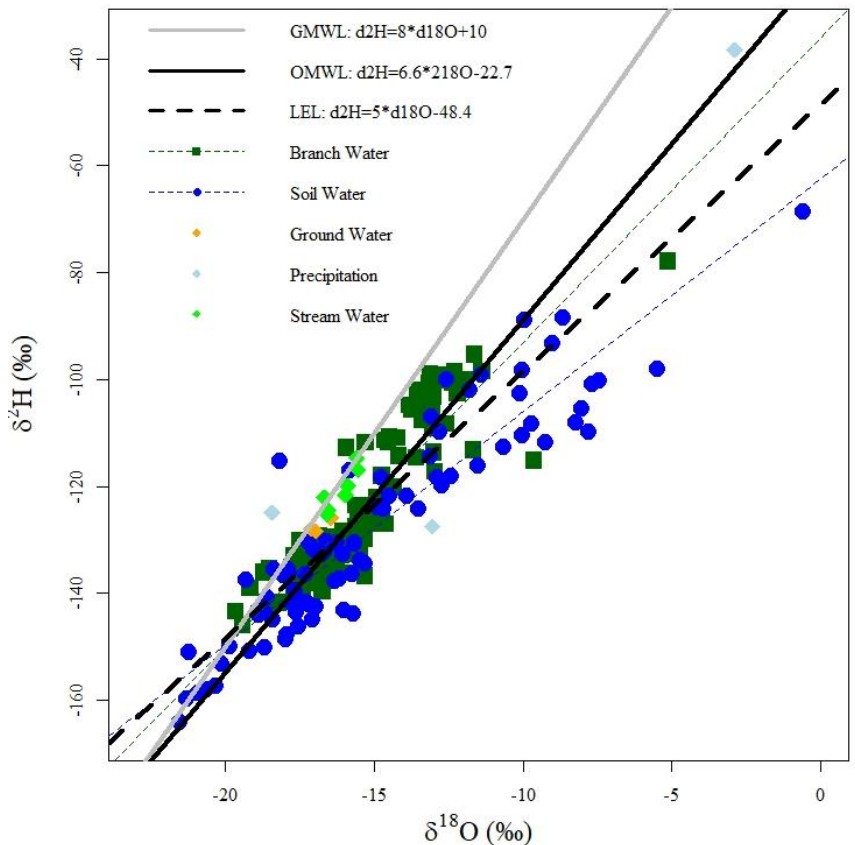

*Figure 6 Biplot of $\delta^{18}O$ and $\delta^2H$ including all branch, soil, stream, groundwater, and precipitation samples collected over the 2021 study period outlined in Table 1 along with the global meteoric water line (GMWL) as well as a meteoric waterline for the Okanagan Valley (OMWL) and the Local Evaporative Line (LEL) developed by Wassnaar et al. (2011). Linear regressions are also plotted for branch water and soil water to indicate deviations in the slope and intercept from OMWL and LEL. The relationship between $\delta^{18}O$ and $\delta^2H$ in soil and branch water shows comparative ranges, but more variation among the soil water samples likely due to changes in precipitation signatures.*

Soil isotopic results were broken into two datasets to analyze the variation in isotopic
composition over time and between treatments, and then a profile of isotopic variance with depth
was constructed. Soil water $\delta^2$H and $\delta^{18}$O varied significantly by depth ($\delta^2$H: p=2.57e-6***;
$\delta^{18}$O: p =2.45e-7***), being higher in the shallow soils than deeper in the profile (Figure 7.A.
and 7.C.). $\delta^2$H varied significantly across months (p=2.72e-5**), but not between July and
September and September and October. $\delta^{18}$O also varied significantly across months (p=1.5e-
5**) except when directly comparing July to October and September to October. Despite
treatment differences in soil moisture (Figure 4), there were no statistically significant treatment
differences in the isotopic composition of soil water at either depth. In June, the mean soil water
$\delta^{18}$O at 5 cm was -16.8±2.57‰ while the
$\delta^2$H was -136.7±13.6‰; at 35 cm, the
$\delta^{18}$O was -19.2±1.52‰ and $\delta^2$H was -
149.2±9.6‰. Both $\delta^{18}$O and $\delta^2$H
increased more during the growing season
at 5 cm than at 35 cm, and with more
variability (Figure 7). In September, $\delta^{18}$O
and $\delta^2$H at 5 cm were -8.75‰ and -106.23
and at 35 cm were -14.71‰ and -127.64
respectively suggesting that soil isotopic
composition nearer the soil surface
follows trends in precipitation samples,
being most enriched with O$^{18}$. By
October, $\delta^{18}$O and $\delta^2$H at 5 cm reflected
more recent precipitation events
indicating that water availability in
shallow soils began to increase.

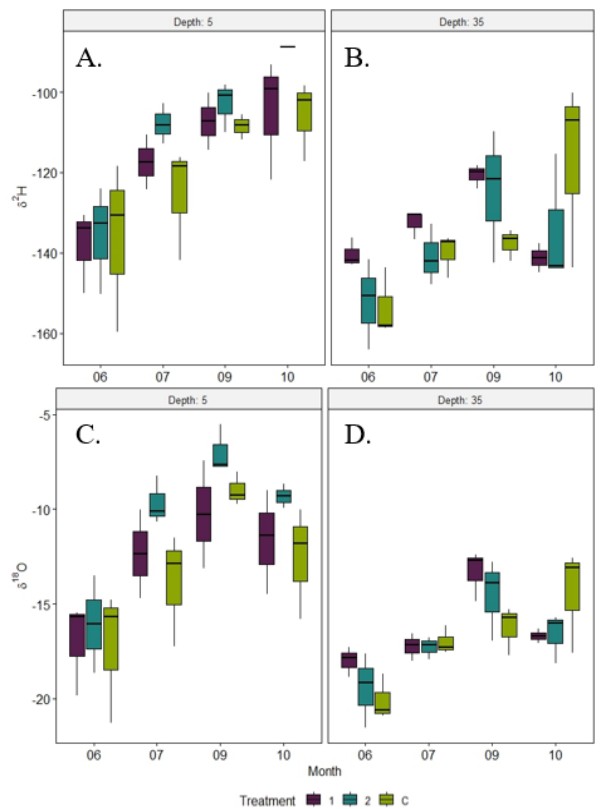

From the isotopic soil profile, there were
three significant groupings of isotopic
composition (p<0.05): shallow soil water
(5-20 cm), deep soil water (35-100 cm),
and groundwater. Mean groundwater
collected at the end of the growing season
most closely resembled spring and fall
snowfall events. The mean $\delta^{18}$O of
groundwater was -16.82±0.34‰, which
resembles that in the soil profile, but
mean $\delta^2$H was slightly higher than soil
water (n=4). This isotope fractionation
may be due to interactions with bound
soil water and soils as the water infiltrates

*Figure 7 Boxplots of the soil water $\delta^2$H (A. and B.) and $\delta^{18}$O (C. and D.) at 5 (A. and C.) and 35 cm (B. and D.) depths collected four times over the growing season from each treatment and block. Mean, interquartile ranges, and standard deviation are indicated for each treatment in each month. There was a significant difference in the isotopic composition of water between months by treatment and depth indicating changes in water isotopic signature either due to evaporation or precipitation.*

through the vadose zone, but the spread of values as potential uptake sources was greater than
any predicted bias from cryogenic vacuum extraction therefor groundwater was included in the
model as a isotopically distinct potential source for lodgepole pine water use (Allen & Kirchner,
2022; Vargas et al., 2017).
The more negative values for both $\delta^{18}O$ and $\delta^2H$ with soil depth indicate that snow melt is the
main source of water to the deep unsaturated zone and that enriched summer precipitation is not
infiltrating deeper soil layers (Figure 8).

3.2.2. Isotopic variability in
branch xylem water
Branch xylem for each
treatment across the three
blocks and the adjacent
mature stand were compared
for each sampling period. All
treatments closely resembled
the mature stand in both $\delta^{18}O$
and $\delta^2H$. There were no
statistically significant
differences in both $\delta^{18}O$ and
$\delta^2H$ of xylem water across
thinning treatments; there
was, however, significant
variation over time ($\delta^{18}O$:
F=24.8*; $\delta^2H$: F=146.6*).
More specifically, $\delta^{18}O$ and
$\delta^2H$ of xylem water varied by
month for all months collected
except for between June and
September and July and

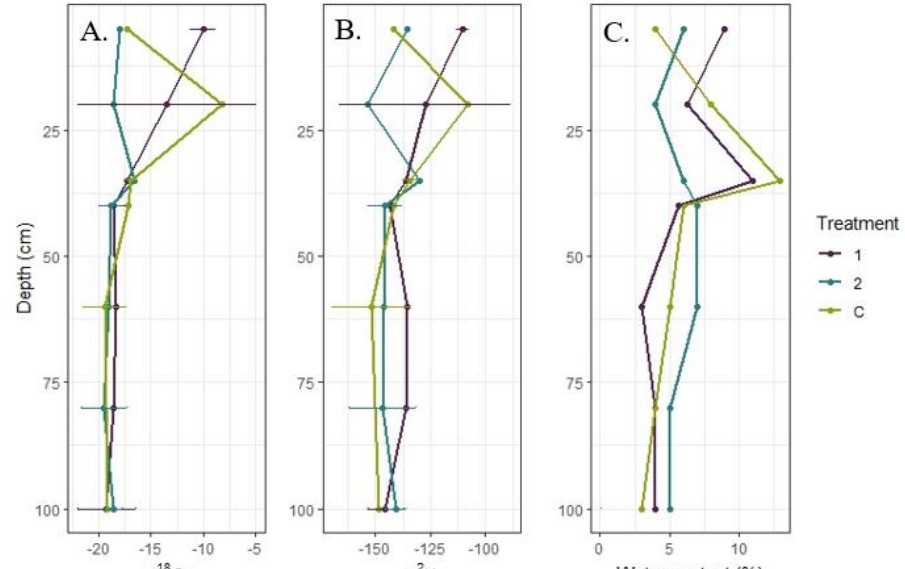

*Figure 8 Vertical isotopic profiles and gravimetric soil water content from treatments in Block 2 and samples collected in mid-July where A. shows the vertical changes in $\delta^{18}O$ for each treatment, B. shows the vertical changes in $\delta^2H$ for ach treatment, and C. shows the change in gravimetric water content as a percent of total soil weight.*

September (Figure 9). Because the isotopic composition of xylem water showed significant
change over the growing season but did not follow the same seasonal trends as soil water, the
trees were likely changing their primary water source within the soil profile.

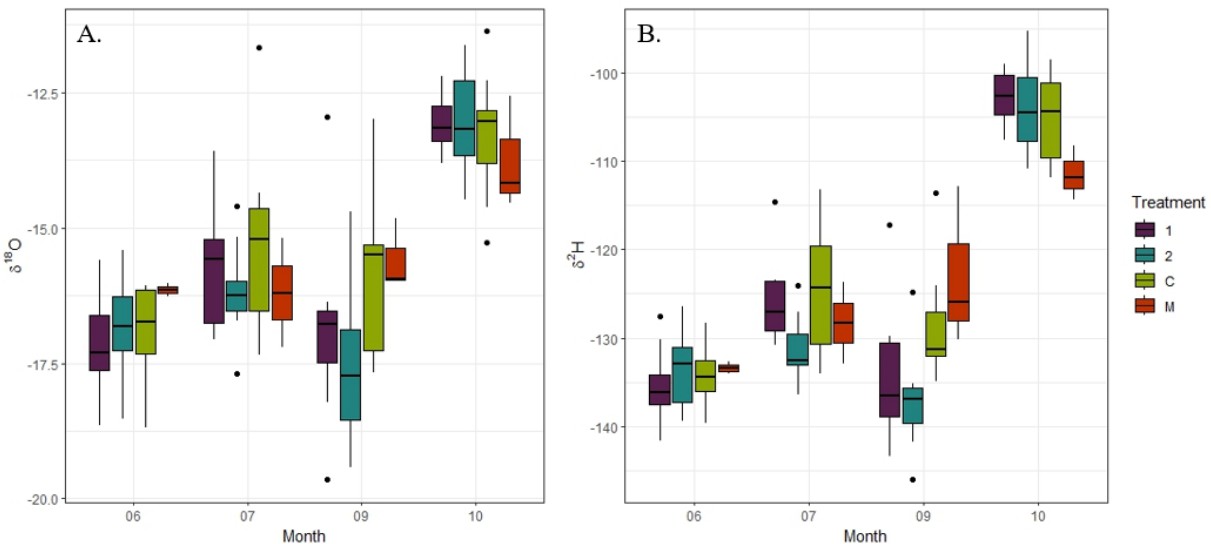

*Figure 9 A boxplot showing branch mean, interquartile range, and standard deviation for A. $\delta^{18}O$ and B. $\delta^2H$ by month and treatments for the control (C), lightly thinned (T1), heavily thinned (T2), and mature (M) stands. Branch water was highest in October despite treatment effects. Mature trees were used as a reference for the isotopic composition of lodgepole pines over time but were not considered in the model of changes depth to water uptake over time. There was not statistically significant difference in $\delta^{18}O$ and $\delta^2H$ between treatments, but each treatment varied significantly by month with the highest concentration of heavy isotopes in October.*

3.3. Partitioning xylem source water and seasonal fluxes using MixSIAR
Of the six scenarios considered, scenarios 1, 2 and 6 approached the Gelman-Rubin diagnostic
(less than 1.05) with a runtime set to "normal" (chain length: 100,000; burn: 50,000; thin: 50;
chains: 3), which indicates that they were the closest of all scenarios to reach convergence (Table
S2). Out of the 6 potential scenarios, scenarios 4 (dual-isotope and 5 cm, 35 cm, and 45-100 cm
soil water as sources) and 6 (duel-isotope 5 cm, 35 cm, 45-100 cm, and groundwater soil water
as sources) were rerun with the run time set to "long" (chain length: 300,000; burn: 200,000;
thin: 100; chains: 3) as they were hypothesized to provide the most representative results of
water uptake partition from various depths. The Gelman-Rubin diagnostic for scenario 4 was
120, and for scenario 6 was 17, when run for the "long" runtime, meaning scenario 6 was closer
to convergence, but still greater than the convergence threshold.
Results of scenario 6 indicate that, in June, trees in each treatment acquired the most water from
the 5 cm depth (C: 76%; T1: 77%; T2: 79%) (Figure 10). In July, shallow soil water was still the
primary source for T1 and T2 at 47% and 61%, but C had 55% water from 45-100 cm deep and
only 33% from 5 cm below the surface. By September, all treatments acquired less than 15% of
tree water from shallow soil. Lodgepole pine water use in treatments 1 and 2 was composed of
approximately 48% and 54% from around 35 cm, while 72% of water in control stand trees was
from 35-100 cm. By October, although SPEI results indicate more moisture and less evaporative
demand, scenario six indicated that all three treatments had most water uptake from below 45 cm
in the soil profile (Figure 10). Results of the MixSIAR model support findings of branch water
stable isotope trends over the growing season where the branch water started with mean $\delta^{18}O$ and
$\delta^2H$ values of -16.9±0.89‰ and -134.37±3.8‰ in June, becoming slightly more enriched in July.

There was a shift to a source with a higher concentration of lighter isotopes in September. Branch water was most enriched with heavy isotopes in October, like shallow soil water, with mean $\delta^{18}O$ and $\delta^2H$ of -12.9±1.76‰ and -103.8±7.0‰, respectively. However, the MixSIAR model does not account for potential changes in the isotopic composition of water from precipitation events from mid-September to mid-October. Additionally, we did not consider extraction bias of the soil water sources nor branch water in the MixSIAR model because the previously mentioned range between distinct sources is larger than the potential change in isotopic signature during cryogenic distillation. The branch water in October was more enriched in heavy oxygen isotopes for each treatment than soil water at a depth of 35 cm and was more isotopically similar to soil water at 5 cm. Deuterium also followed a similar trend.

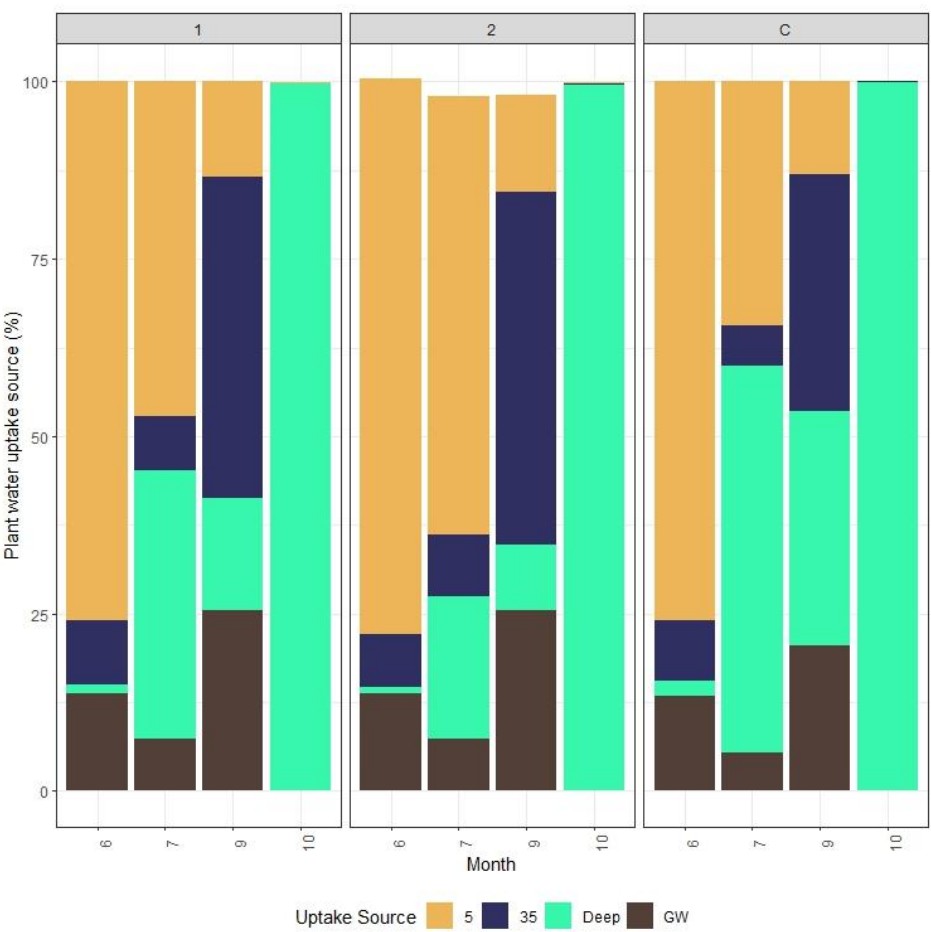

*Figure 10 Stacked bar charts showing the partitioned relative contribution of different sources of water in the soil profile by the MixSIAR model of scenario 6 with long (chain length: 300,000; burn: 200,000; thin: 100; chains: 3) runtime. Scenario 6 considers both $\delta^{18}O$ and $\delta^2H$ as tracers and 5 cm, 35 cm, 45-100 cm, and groundwater as potential sources. Soil water isotopic composition at 5 cm and 35 cm changes monthly whereas the concentrations of $\delta^{18}O$ and $\delta^2H$ are held constant for the 45-100 cm and groundwater sources. Results of this model indicate that there are significant changes in the depth to water uptake of lodgepole pines between June and October of 2021 and that thinned trees can maintain a larger percentage of water uptake from shallow soil water longer than trees in the control stand.*


4. Discussion
4.1. Seasonal variability in soil water
Deep soil water showed mixed gradient of older, more depleted, water molecules deeper in the
profile indicating that deep soil water mainly originates from spring snowmelt during the
summer months. Low intensity and less frequent summer precipitation events are evaporated out
of the shallow soil layers and do not infiltrate past the evaporative front to recharge the
unsaturated zone or groundwater. Although there was not a statistically significant difference in
the depth to water uptake by thinning treatments, the results of our isotopic analysis indicate that
there was increased evaporative enrichment, or a higher concentration of oxygen-18, in the
shallow soils of the heavily thinned stand compared to the oxygen-18 concentrations in the
moderately thinned and control stands (Figure 7.C.). The muted enrichment in $^{18}$O around 35 cm
depth in the soil indicates a mixing of the left-over summer precipitation with older and lighter
water. Our results do not indicate that differences in soil exposure canopy coverage were
effective enough to significantly affect the isotopic composition of soil water below 5 cm in
depth.
4.2. Seasonal lodgepole pine water use
Literature utilizing stable water isotopic analysis to determine plant preferential water uptake in
arid regions indicates that vegetation can utilize precipitation despite the temporal origin
(Andrews et al., 2012; Brinkmann et al., 2019; Ehleringer et al., 1991). Seasonal water
availability depends on precipitation, soil water holding capacity and drainage, and evaporative
loss (Gibson & Edwards, 2002; Kleine et al., 2020; Stumpp et al., 2018). Based on the seasonal
shift in the isotopic composition, soil water at a depth of 5 cm was more enriched with heavier
isotopes over the growing season than at 35 cm due to more evaporative isotopic fractionation
near the soil surface and a lack of rainfall intense enough to drive precipitation deeper into the
soil profile before September 16, 2021 (Figure 3). The effect of evaporative enrichment of the
near surface soil water was most obvious in July and September in the heavily thinned stand
(T2). However, variability in branch isotopic composition did not follow the same trends. Our
results indicate that lodgepole pines access water from multiple depths in the soil profile.
Regardless of depth and forest density, spring snowmelt is the main source for lodgepole pines as
it infiltrates through the vadose zone.
The MixSIAR isotopic partitioning model results from each of the six scenarios indicated a
seasonal shift in the depth to water uptake of lodgepole pine, regardless of changes in stem
density, over the growing season. At the beginning of the growing season, when snow meltwater
is more available at shallow depths and beginning to infiltrate through the soils, lodgepole pines
obtain most of their water from snow melt in shallow soils with small contributions from other
potential sources (< 25% of June water uptake in all treatments). Then, in July, the trees in the
control treatment were using less shallow soil water (34.3% of plant water uptake from 5 cm
below the soil profile) whereas the moderately thinned and heavily thinned plots maintained a
greater proportion of shallow water uptake (47.1% and 61.5% respectively). The mean $\delta^{18}$O and
$\delta^{2}$H of branch water from each treatment in September had a higher concentration of lighter
stable water isotopes than in July and a larger proportion of tree water was from 35-100 cm deep
in the soil profile as shallow soils were dry from a lack of rainfall and surface soil evaporation.
By September, the control stand was more dependent on deeper soil water and groundwater with
only 33.4% of plant water uptake originating from 35 cm in the soil profile, whereas both
thinning treatments maintained more than 45% of water uptake from 35 cm in the soil profile. In
October, all treatments were completely dependent on deep soil water, but it is likely that the
isotopic profile of deep soil water sampled in July skewed the results. It is plausible that the trees
began to rely on shallow soil water towards the end of the growing season when soil water
content increased. Further research is needed with more intensive sampling of deep soil water
during the hydrological recharge period at the end of the growing season and beginning of
senescence.
Local monitoring close to the study site indicated that the depth to groundwater stayed at least
6.5 m below the surface from August through the end of the study period. The continued use of
deep soil water even during rewetting in late September and October suggests that the drought
conditions suppressed top soil water uptake, but that deeper soil was sufficiently saturated to
sustain root water uptake and tree function enough to limit groundwater uptake to less than 30%
for all treatments until the beginning of fall precipitation events recharging the saturated zone.
Our results indicate that lodgepole pine, like other pine species in arid regions, is flexible in its
ability to access deep soil water and can change its depth to water uptake over time depending on
water availability (Brinkmann et al., 2018; Grossiord et al., 2017; Kerhoulas et al., 2013; Kleine
et al., 2020; Moreno-Gutiérrez et al., 2011; Simonin et al., 2006; Sohn et al., 2014; Wang et al.,
2021). Our results of seasonal changes in depth to water uptake by lodgepole pine support the
findings of Andrews et al. (2012) on changes in lodgepole pine depth to water uptake in Alberta.
Tree species native to arid regions exhibit a variety of adaptations to long-term drought stress
and decreased water availability in the soil profile such as deep tap roots, access to the water
table, utilizing bound and mobile soil water, fine root mortality, and hydraulic redistribution in
ecosystems with low water holding capacity (Amin et al., 2020; Brinkmann et al., 2018;
Grossiord et al., 2017; Kerhoulas et al., 2013; Kleine et al., 2020; Langs et al., 2020; Meinzer et
al., 2007b; Prieto et al., 2012; Sohn et al., 2016; J. Wang et al., 2017, p. 201).
The literature is inconsistent across different biogeoclimatic regions and species with regards to
the effects of thinning on stand dynamics that influence inter-tree competition for water
resources or changes in depth to water uptake. (Kerhoulas et al., 2013; Moreno-Gutiérrez et al.,
2011; Sohn et al., 2016; Wang et al., 2021). We found no significant impact of forest thinning on
depth to water uptake. However, our observation of seasonal shifts in depth to water uptake
support results of a study on the impacts of thinning intensity on 60-year-old *Pinus halepensis*
Mill. in a semi-arid region of Spain which concluded that forest thinning reduced competition for
water resources but did not alter water uptake patterns (Moreno-Gutiérrez et al., 2011). Another
study on the impact of thinning *Pinus ponderosa* Dougl. on depth to water uptake concluded that
water was consistently more isotopically enriched in low-density stands potentially due to
prolonged evaporative fractionation in the soil profile, or that understory vegetation utilized
more shallow water sources (Kerhoulas et al., 2013). The impact of forest thinning on stand and
understory water use is highly variable and dependent on understory growth, canopy structure,
water availability, when forest thinning is implemented, and the time since stem removal
(Kerhoulas et al., 2013; Moreno-Gutiérrez et al., 2011; Sohn et al., 2016). More research is
needed to decern if lodgepole pine relies more on mobile or bound soil water, the extent of
lodgepole pine rooting zones, what biogeochemical factors cause seasonal shifts in water uptake,
and if severe seasonal drought has a lasting effect on water uptake strategies during hydrologic
recovery (Simonin et al., 2007; Vargas et al., 2017).
4.3. Impacts of the drought and implications for future climate conditions
The 2021 growing season was an abnormally hot and dry period for the interior of British
Columbia with severe to exceptional drought conditions. Wang et al. (2019) found that thinning
improved water-use efficiency, drought tolerance, and drought recovery by decreasing stand
density and improving carbon storage. Our results support the finding that lodgepole pine trees
can adjust to prolonged water scarcity, and over-populated stands may be more resilient than the
literature has initially indicated. In fact, drought conditions over the study period likely
intensified the change in xylem water isotopic composition over the growing season. However,
the scope of this study did not include pre-drought seasonal water use patterns nor the impact of
forest density on depth to water uptake during drought recovery. Because lodgepole pine depth
to water uptake changes during prolonged dry growing season conditions, the trees are more
reliant on winter snowpack and spring infiltration to recharge deeper source water below the
evaporative front. One experiment on juniper (*Juniperus monosperma* (Engelm.) Sarg.) and
pinion pine (*Pinus edulis* Engelm.) investigated the simultaneous stress of increased heat and
decreased precipitation on depth to water uptake and found that extreme temperatures and
decreased precipitation lead to less reversable embolism and more root death in surface soil
levels preventing trees from accessing shallow water sources if precipitation becomes more
available late in the growing season (Grossiord et al., 2017). It is becoming more imperative to
understand the climatic drivers of lodgepole pine water use and access as mean annual
temperatures continue to rise, the seasonal frequency and intensity of precipitation change, and
drought conditions become more severe. This study indicates that severe seasonal dryness pushes
lodgepole pines to rely more on snowmelt while losing function in shallow roots. Our results are
inconclusive in determining the depth to water uptake in September and October because of
limited deep soil water measurements. However, increased annual temperatures and more
variable precipitation patterns as a part of climate change projections are predicted to drive
decreases in winter snowpack and could drive lodgepole pine stands, regardless of stem density,
to rely on groundwater influencing water availability and depth to groundwater. These
projections could lead to prolonged inter-annual water scarcity along with seasonal water
scarcity during the late growing season.

5. Conclusions
Lodgepole pine, across all treatments, was able to shift access from shallow soil water at the
beginning of the growing season to deeper soil water as drought conditions intensified. The
quick-draining and sun-exposed soils of the UPC do not retain small summer precipitation
events, and these patterns are intensified in the shallow soil layer of the heavily thinned stand
because decreased canopy cover can be directly related to increased soil evaporation. As a result,
due to changes in water availability, lodgepole pines shift to a more readily available source in
the soil profile (Aranda et al., 2012; Prieto et al., 2012). Our findings support the literature that
lodgepole pines are a drought-tolerant species with dimorphic rooting systems allowing them to
access water from varying depths in the soil depending on water availability (Andrews et al.,
2012; Liu et al., 2011). Despite the ecological plasticity under extreme heat and low summer
precipitation conditions, there was no statistically significant variance in depth to water use
between the over-populated plots and thinned ones. Both thinned and unthinned lodgepole pine
stands were able to access shallow soil water during the early months (June and July), then
switched to deeper soil water and a larger proportion of groundwater during September.
Although there was not a statistically significant difference in isotopic composition of branch
water for the different treatments, our results indicate that decreased stem density may lead to the
prolonged use of soil water 35 cm below the surface during prolonged dry periods which would
decrease the dependency of lodgepole pine on shallow soil water and summer precipitation
events and rather increase the dependency on deep soil water or ground water fed by winter snow
accumulation and spring snowmelt.
Future climate projections indicate hotter growing seasons and less precipitation (Allen et al.,
2010). Further investigation is needed to discern how lodgepole pines, under different stand
densities, use water during prolonged drought and drought recovery periods (Grossiord et al.,
2017; Navarro-Cerrillo et al., 2019; Simonin et al., 2007; Sohn et al., 2016). From our findings,
stand density did not prevent lodgepole pines from accessing soil water from various depths, but
decreased stem density did result in lodgepole pines using soil water higher in the soil profile for
longer under extremely dry conditions. Lodgepole pines indicate a strong level of drought
tolerance and ability to access water under extreme heat conditions. If summer precipitation
decreases, lodgepole pine in the interior of British Columbia can access deeper soil water from
spring snowmelt. However, if snowpack and spring snowmelt begin to decrease, lodgepole pine
may need to acclimate to these hydrological shifts.

*Code and Data Availability:*
The codes of the data analysis and plotting are available at https://github.com/emory-
ce/LodgepolePineWaterUseStrategies2021 and are available upon request (ece58@nau.edu)

*Author Contributions:*
EE conceived the idea as a part of their Master's research with AW, and performed the
extractions with RG. Analysis was primarily conducted by EE with guidance from AW and RG.
All authors contributed to the manuscript.

*Competing Interests:*
None of the authors have competing interests.

*Acknowledgements:*
This study was funded by the Ministry of Forests, Lands, Natural Resource Operations and Rural
Development. Field work was done with the assistance of Fiona Moodie. Cryogenic distillation
was conducted at the University of British Columbia. Samples were sent to the Stable Isotope
Facility at University of California, Davis.

*Financial Support:*
This research was funded by the Ministry of Forests, Lands, Natural Resource Operations and
Rural Development (grant number: RE21NOR-029)

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
