# Peer review of "Seasonal shifts in depth to water uptake by young thinned and overstocked lodgepole pine"

_EGUsphere, 2024_

## Author Comment (AC1)

New Figures

Continuous Soil Moisture:

[Figure]

Revised Soil Depth Profile

---

## Author Response (AR1)

Responses to comments from the Editor and reviewers

\*\*\*\*\*\*\*\*\*\*\*\*\*\*\*\*\*\*\*\*\*\*\*\*\*\*\*\*\*\*\*\*\*\*\*\*\*\*\*\*\*\*\*\*\*\*\*\*\*\*\*\*\*\*\*\*\*\*\*\*\*\*\*\*\*\*\*\*\*\*\*\*\*\*\*\*\*\*\*\*\*\*\*\*\*\*\*\*

Editor

Please, address all Referees' comment and try to integrate, as much as possible, results and discussion with additional data on water balance and interaction with atmosphere. The revised manuscript will be sent out for further revision.

Dear editor,

Thank you for the opportunity to additional data and integrating the comments of the reviewers. We have also addressed all of the feedback in our manuscript as described below.

\*\*\*\*\*\*\*\*\*\*\*\*\*\*\*\*\*\*\*\*\*\*\*\*\*\*\*\*\*\*\*\*\*\*\*\*\*\*\*\*\*\*\*\*\*\*\*\*\*\*\*\*\*\*\*\*\*\*\*\*\*\*\*\*\*\*\*\*\*\*\*\*\*\*\*\*\*\*\*\*\*\*\*\*\*

**Reviewer #1**

I think that the data presented in this paper may allow for a more nuanced message regarding the impact of the thinning treatments on the water use pattern and uptake depths of lodgepole pine in overstocked stands during drought. The data presented in Figure 7 appear to indicate a faster and earlier depletion of topsoil water storage (5-35 cm depth) by trees in the overstocked control stands during late spring and early summer, as well as an earlier shift to deeper water sources utilization (>35 cm depth). In contrast, trees in the thinned stands (T1, T2) continue using a much larger proportion of topsoil water further into the summer, and shift to uptake of deeper water sources later in the summer season. This pattern appears to be particularly evident for the data collected in July, when trees in the unthinned control treatment were already obtaining more than 60% of their water from deeper sources (>35 cm), whereas trees in the thinned T1 and T2 treatments we still obtaining 55-70% of their total water uptake from shallower soil layers (5-35 cm depth). This pattern strongly suggests to me that the thinning treatments were rather effective at delaying the consumption and depletion of the available topsoil (5-35 cm) water pool by the pines during the early stages of the summer drought, compared to the overstocked control stands.  In other words, trees in the unthinned control stands were forced to shift to deeper soil water sources earlier than trees in the thinned stands, probably due to faster depletion of the topsoil water pool by the overstocked control stands.   I agree with the authors that the water uptake depth of the lodgepole pines was less affected by the thinning treatments later into the summer drought period, although in August the trees in the thinned stands were still using a somewhat greater proportion of topsoil water (5-35 cm) than the unthinned control stands.  So I think that the main message highlighted in the summary and conclusions sections ("forest thinning did not cause a significant change in depth to water uptake") is questionable and should be reconsidered and/or significantly modulated after careful re-analysis and re-interpretation of the data.

Response: Thank you for your positive feedback and contribution to the interpretation of our findings. In response to your general comments, we agree that the results shown in our final figure indicate that the trees in the control stand show less dependence on shallow soil water, specifically when compared to the heavily thinned treatment, and this trend is most pronounced in July. We re-examined the results of the mixing model and revised section 3.3 (Partitioning xylem source water and seasonal fluxes using MixSIAR) to have a more detailed explanation of the patterns over the growing season, with special note of fluxes between July and September for each treatment (lines 485-491). What we found is in agreement with your overarching comment that in July, trees in the control treatment were using less shallow soil water than the two thinned treatments (34% for the control treatment, 47% for T1, and 61% for T2). In September the three treatment had similar proportions of water use from the shallow soil water,  but the control stand was more dependent on groundwater and deep water where as the thinned treatments were still able to get more than 45% of their water from 35 cm.  These more detailed findings are outlined in our abstract (lines 29-34), discussion (lines 505-508 & 512-515), and conclusion (lines 603-606 & 610-613).

September, trees in the more thinned treatments were able to use soil water from below the evaporative front more than those in the control stands which had a larger proportion of soil water uptake from deeper in the soil profile.

Also, I would like to see a much more explicit and detailed description of the impacts of the thinning treatments on the continuously measured soil moisture content data (L311-313). I would even recommend including a new figure to better highlight any differences encountered in soil moisture contents by depth among the different treatments/stand densities.

Response: Thank you for the suggestions. We added an additional information on the soil moisture sensors that we used (lines 176-178) in our methods section along with plots and an interpretation in our results section (lines 318-331).

The data presented in Table 1 appear to indicate greater rainfall interception by the tree canopy during late summer/early fall in the overstocked control stands than in the thinned stands, as suggested by the large differences in topsoil water contents (5 cm) at the end of the growing season. Also, please clarify whether this table reports data measured in October, and please note that the standard deviations mentioned in the figure caption are actually missing from all the columns, so please include them. The data presented in this table appear to suggest that mean soil water content across the full soil profile (5-100 cm depth) was slightly higher in the T2 treatment than in the unthinned control treatment at the end of the growing season, which would be a relevant finding if confirmed by appropriate statistical analyses.

Response: We did not directly measure canopy cover nor interception, but the additional figure of continuous soil moisture (Figure 4) supports your comment and the results in Table 1 that there was more throughfall in the thinned stands. Because those results are now shown in Figure 4 and the rest of the information from the table (previously table 1) of mean d18O, d2H, and volumetric soil water content can be found in Figure 8, we opted to remove Table 1 from the manuscript. We did test to see if the soil water content of T2 was larger than the other treatments, but there was not a statistically significant difference, so we did not add this to our results.

In figure 5, it is unclear why data are missing for the upper soil layers in some of the treatments, or why the scale of the Y axis is not uniform across panels. In Figure 7, it is unclear why the values in the Y axis range between 0-1 instead of 0-100, as the axis title refers to percentages. It is also unclear why some of the columns in the central panel (T2) are shorter than all the other columns in the figure.

Response: In our latest revision, Figure 5 is now Figure 8 (line 425). Thank you for noticing that the 5 cm depth data was missing from some of the panels. This has been corrected in this version. As for Figure 10 (previously Figure 7; line 452), we corrected the y axis scale to range from 0 to 100 to correspond with the axis title. The mean relative proportion of water uptake by depth did not add up to 1 in the T2 panel in July or September, but did account for over 100% because, as the Gelman-Rubin diagnostics indicated, the model did not converge for these two instances. Even still, our model with the four different potential water sources provided the best overall assessment of the proportion of water taken from each depth.

L453-455: Several studies have shown that trees that have access to relatively shallow groundwater (6.5 m deep at the study site) can support the integrity and functionality of their shallow fine roots and associated mycorrhizal fungi in dry topsoil layers during prolonged drought through internal hydraulic redistribution (Bauerle et al 2016; Querejeta et al 2007), so the assumption made in L454-455 and L508 is perhaps speculative and questionable, in the absence of any direct measurement of root function. Are there any other plausible explanations for the apparent inability of the lodgepole pines to use recent rainwater during the late growing season? Perhaps those late season rainfall events were of insufficient magnitude to recharge the topsoil layer in a physiologically meaningful way? In L510, do you mean that pines are unable to access water made available by late-season rainfalls during the rewetting period?

Response: We made adjustments in both our introduction and discussion to remove any mention of rooting depth/ function in relation to our results because, as mentioned in your comment, we did not take any direction measurements of root function. The paragraph (lines 521-535) pulls our data on soil moisture (from the continuous soil moisture results we added) and the results of the MixSIAR model to discuss suppressed soil water uptake in the shallow region and sufficient soil water availability from deeper layers in September to support plant water uptake to keep groundwater uptake less than 30% rather than speculating on fine root mortality (lines 531-535).

I think that the strong assertions made in L520-522 and 526-528 of the Conclusions section (and similar statements in the Abstract) should be reconsidered and rewritten after careful examination of the issues raised above.

Response: We reconsidered our results and although there is not a statistically significant impact of thinning on depth to water uptake, we provide a more nuanced conclusion on the proportion and timing of shifts in water uptake for each treatment (lines 607-611).

Some of the references mentioned in the text appear to refer to wet riparian habitats and tree species, and I thus wonder whether they are really directly relevant to this particular study which has a strong focus on dry interior forests of the semiarid Okanagan Valley (e.g. Gibson &Edwards 2002; Liu et al 2015; Maier et al 2019; Sanchez-Perez et al 2008). It might be more appropriate to replace these references by others referring to semiarid pine forests more closely resembling your study system.

Response: We opted to leave in the citations by Lin & Sternberg, 1993, Maier et al., 2019, Sanchez-Perez et al., 2008, and Liu et al., 2015 as they were used in the introduction to establish the breath of current literature reporting depth to water uptake of various species and levels of water availability. However, we did remove the citation by Gibson & Edwards (2002) as it was not relevant to the sentence and, as you mentioned, not an appropriate ecosystem to draw parallels between. The text was also adjusted.

L30-31 in Abstract: They cannot shift from deep water sources when shallow water becomes more available at the end of the growing season? Or perhaps they don't need to shift to shallow water because deep water is plentiful and sufficient to meet their transpiration demand (which is presumably lower in October than in midsummer)? Or perhaps these late-season rain events are

insufficient to recharge the topsoil layer meaningfully (i.e. the soil matric potential in upper layers may still remain lower than that in deeper layers in October despite these rain events)?

Response: We clarified our conclusion in lines 31-34 that lodgepole pines do not shift back from deep water sources when soil water availability increased following precipitation events at the end of our study period and removed "cannot" which we believed to lack clarity and leave speculation as to changes in root function limiting where trees accessed water sources.

BAUERLE, T.L., RICHARDS, J.H., SMART, D.R. and EISSENSTAT, D.M. (2008), Importance of internal hydraulic redistribution for prolonging the lifespan of roots in dry soil. Plant, Cell & Environment, 31: 177-186.

Querejeta JI, Egerton-Warburton LM, Allen MF (2007) Hydraulic lift may buffer rhizosphere hyphae against the negative effects of severe soil drying in a California Oak savanna. *Soil Biology and Biochemistry*, 39, 409–417.

Response: We opted not to include these citations as we tried to restructure our paper to no discuss root function as much.

\*\*\*\*\*\*\*\*\*\*\*\*\*\*\*\*\*\*\*\*\*\*\*\*\*\*\*\*\*\*\*\*\*\*\*\*\*\*\*\*\*\*\*\*\*\*\*\*\*\*\*\*\*\*\*\*\*\*\*\*\*\*\*\*\*\*\*\*\*\*\*\*\*\*\*\*\*\*\*\*\*\*\*\*\*\*\*\*\*

Reviewer #2

The study by Ellis investigates the impact of thinning and seasonal water availability on root water uptake of lodgepole pine trees in an experimental forest in Canada using water stable isotopes. The authors found that the root water uptake of pine trees responded to the seasonal shift in water availability but not to thinning treatment. Overall, the findings are timely and important for similar research in the field, as well as nicely presented and discussed. My comments probably lead only to minor modifications of the manuscript.

Response: Thank you for your positive and informative feedback. We have carefully considered your comments and incorporated them into our most recent draft.

Title: The title of the manuscript could be a bit more specific, so that it matches the story of abstract, and the main findings of the manuscript. For instance, it's not 100% clear what "unexpected" means throughout the manuscript and whether it's related to the effect of soil water availability or thinning. Maybe add "Canada" to the title.

Response: We agree with your comment that the initial title lacked clarity and have changed the title to "Seasonal shifts in depth to water uptake by young thinned and overstocked lodgepole pine (Pinus contorta) forests under drought conditions in the Okanagan Valley, British Columbia, Canada" which gives a better description of our manuscript findings as well as the geographic location of our research.

Line 158-160: Font and/or font size changed. Please check.

Response: We diligently revised the entire document for formatting issues such as test font, size, and color.

Line 177-183: Indicate briefly how close the soil pits were to the "sample" trees? What was the exact sampling period for the vertical 1 m pit samples. Where the soil sample taken directly after digging the pits?

Response: The location of our soil pits varied of the sampling campaign to ensure that we were sampling undisturbed soils, but we added detail to the description of how we sampled soils (lines 193-202)

Line 196-198: The process of sampling handling of soil and plant samples between sampling in the field and before CVD extraction is not clear. For instance, were the samples transferred into a new tube or glass vial before CVD? What is a test tube?

Response: We added additional details on the management of samples, materials used to store samples, and preparation before CVD throughout section 2.3. Sample collection.

Line 196-208: CVD extraction can affect isotope values of the extracted water (Chen et al, 2020, PNAS, Barbeta et al 2022, New Phytologist). Given the growing number of papers stating a bias for d2H values of woody material, I was wondering whether such a bias could affect the conclusion of the manuscript. This is because d2H values are used in the MixSIAR model to estimate root water uptake. Please clarify. Please also briefly give a statement on the extracted water amount (Diao et al. 2022, HESS).

Response: This is excellent feedback as we had not discussed extraction bias in our initial submission. We added additional information in our methods section (lines 235-237) and using the findings of Allen & Kirchner (2022) which found that there is a 4-6‰ bias in the $\delta^2H$ extracted using CVD and all of our potential sources were distinctly different by more than 4‰. Additionally, all of our water samples were extracted using the same vacuum distillation setup and methods which limits bias. Because our results are comparing samples to each other rather than the exact value we feel as though our additional discussion of extraction bias is sufficient to justify our results. As for extracted water volumes we added that information into line 238.

Line 249: But samples were taken every 20 cm, not every 10 cm, right? See Line 181-182.

Response: This was an oversight in our initial submission and has been corrected.

Line 293-301 and Figure 3: Some results are not well described. For example, lines 293-295: which samples exactly were used to determine the slope and intercept? Additionally, there is a lack of description regarding branch, soil, ground, and stem water. While OMWL is a bit unusual, it is acceptable. What does LEL represent? Furthermore, distinguishing between blue and green values is difficult in the small figure. Could one 'zoom in' or alter the colors slightly to ensure differentiation, such as between branch and soil water isotope values?

Response: We agree that our initial explanation of the OMWL and LEL was lacking and lead to confusion in the biplot. We added addition information on the trends in our branch and soil water that was sampled and how it falls on the biplot in relation to OMWL, GMWL, and trends line for the samples (lines 352-360). We also enlarged the figure and made the individual points larger to improve the readability.

Table 1: Standard deviation not shown as stated in the caption. What does SMC mean?

Response: We removed this table as it did not add additional information that was not already available in figure 8 showing the vertical soil profile.

Line 367 and throughout: d2H and d18O can be higher or lower, but not depleted or enriched. However, e.g., soil water can be 18O-depleted or 2H-enriched.

Response: We changed the wording to be more accurate throughout the manuscript

Line 369 and throughout: Please consider placing "isotope" before the word "fractionation" to make more explicitly what is actually fractionated.

Response: We took this under consideration and added the word isotope in front of fractionation when appropriate throughout the document.

Figure 7: In contrast to 5 and 35 cm soil samples, "deep" soil water has been sampled only once per growing season. Therefore, the deep soil samples lack a temporal component, right? Is the potential variation in deep soil water isotope values with time irrelevant for the study conclusion? If yes, can the authors back this up for the experimental site? Would it make sense to consider precipitation (e.g. modelled precipitation) as an additional source or do the authors assume that the 5 cm soil samples reflect isotope variation in precipitation?

Response: Correct, deep soil water does not have temporal variability. With the addition of precipitation and continuous soil moisture data, it is plausible that deep soil water in October was not isotopically similar to when we sampled below 35 cm in July because of increased rainfall frequency and intensity. Because of this we added this potential source of error in our results to the discussion section (lines 532-535).

Discussion point 4.2: Does the soil evaporative effect, regardless of transpiration, increase with increasing thinning?

Response: We did not quantify the depth in the evaporative front over the growing season, but figure 7 does indicate that the water was more enriched in heavy isotopes during the late summer months. This may be either due to precipitation events with a higher concentration of heavy isotopes or due to a higher evaporative intensity in the shallow soils. We did add this to our discussion (line 511)

Line 471: Sentence not clear, please rephrase.

Response: We revised this sentence and other unclear sentences throughout our paper.

\*\*\*\*\*\*\*\*\*\*\*\*\*\*\*\*\*\*\*\*\*\*\*\*\*\*\*\*\*\*\*\*\*\*\*\*\*\*\*\*\*\*\*\*\*\*\*\*\*\*\*\*\*\*\*\*\*\*\*\*\*\*\*\*\*\*\*\*\*\*\*\*\*\*\*\*\*\*\*\*\*\*\*\*\*\*\*\*

Reviewer #3

The manuscript submitted by Ellis et al. approaches a timely and very interesting topic especially during intensified climate change conditions. However, I feel it does not qualify to be published at this stage. And I do say this with regret because I know the first author is an ECS and this work the result of a master thesis. Therefore, my critique is primarily directed towards the more senior authors. It's crucial for senior researchers to ensure the readiness of their work before submission, and this feedback is not intended to deter the young researcher's enthusiasm or efforts, because this work has been a great effort as a master thesis.

Water stable isotopes are wonderful tools for supporting inferences about flux origins and necessary to support research in forest hydrology or physiology. However, they often cannot stand alone without additional information about the fluxes of interest, and this is, in my opinion one of these cases. The authors monitor soil and branch water stable isotopes throughout an entire season without adding information about precipitation amount and transpiration fluxes, but because 2021 seemed to be an exceptionally dry year in Canada, this would have been key. Inferring water uptake depths without knowing how much water left the system in which direction (i.e. as transpiration or percolation) is not meaningful and results in the discussion turning around speculations. I also read the publication by Wang et al. 2019, which the authors mention, and it seems that the differences in transpiration fluxes are significant especially in the drier year of 2017. This information would greatly support parts of the current manuscript but is not provided. If transpiration in the control stand is as significantly reduced as shown by Wang et al. for 2017, then how can you be sure that there are significant amounts of transpiration in the even drier 2021, especially considering that you describe scorching in the crown? Subsequently the conclusion you draw that forest thinning does not significantly influence the change in water uptake depth is misleading and at least incomplete because likely the transpiration in the control plot is so low that no significant amounts of water do leave the system.

Response: Thank you for your thorough response, we have carefully taken your comments under consideration and believe that our revised submission is restructured in a way that addresses our question in a more organized and logic way. We added continuous soil moisture and climate data during our study period that better supports our findings on the temporal variance in depth to water uptake and removed much of the speculation about lodgepole pine root function under drought conditions. We also referenced the publication by Wang et al. more directly in the introduction to introduce (lines 102- 106 and 117) the known mechanics of our study site. We also removed some of our observations that were not directly measured or supported in our initial manuscript. Most significantly, we changed our initial hypotheses to fit what we tested in this study and conclusion to be less on stand density and competition but more directly connected to seasonal changes in depth to water uptake of lodgepole pines at different stand densities.

In case you do decide to move forward in finding a way to provide this information or re-writing the manuscript with a different research question, I do have some more specific comments that can help moving forward:

INTRODUCTION:

L 111-113: why would they be limited in depth? Would not the horizontal extend be limited in an overstocked forest plot?

Response: Because we did not measure root density at different depths, we removed this from our hypotheses because it was misleading. Instead, we hypothesize that at lower densities trees could maintain mid-level soil water uptake due to reduced competition (lines 125-127).

METHODS:

L 123: course = coarse?

Response: Yes

LL 149-152: This study should already be mentioned in the introduction as it leads to the formulation of hypothesis 1.

Response: The Wang et al. article has been moved to lines 117-121 to set a framework for our hypotheses.

General comment: this section would benefit from a tabular overview of the sampling campaigns for all the compartments. It is very difficult to follow when you sampled what and a table could help solve this issue.

Response: We added Table 1 (line191) with details of our sampling campaign.

L 180: "in the middle of the growing season…" when is that exactly, please provide the dates to when you did things.

Response: We clearly defined the dates of collecting 1 m pit soil samples (line 197)

LL 184-193: "Precipitation samples were collected when available during field collection days." What does this mean? Is this then a cumulated precipitation sample? "Groundwater and stream samples were collected at the end of the growing season as stream beds were dry and groundwater was inaccessible during the dry period." When was this? Please provide the dates. Also see above, fill a table with this information so that the reader knows when you have which data available. From how this reads you have three snow/rain samples and samples groundwater once?

Response: In lines 203-206 we clarify the cumulative precipitation sampled collected across rain events that occurred during field work. The date of precipitation events are also in Table 1.

L 220: give more info on the LEL

Response: We define the local evaporative line more clearly in line 255-259 as well as in our results.

RESULTS:

Figure 3: Please change the colours to more distinct and enlarge the text and axis descriptions (in R at least size=16)

Response: We enlarged the biplot (now figure 6 line 360)

Table 1: please add the SD's to the table

Response: We removed this table as it was not adding to the manuscript and all the information from the table is also available in Figure 8 (line 419).

Figure 5: why is the uppermost soil layer missing for some treatments? Please make sure you also mention this in the text.

Response: This was an oversight and has been corrected with the complete figure including all data points.

L369: could also be cryogenic extraction bias? See (Allen and Kirchner 2022)

Response: We added discussion of cryogenic extraction bias as mentioned by Allen and Kirchner (2022) in our methods section (lines 235-237) and discussion of the MixSIAR model results.

Figure 6: What is the M treatment? Also the figure caption is incomplete. What do the boxes show? Again also enlarge axis and figure text.

Response: We improved this figure to be consistent with our other figures throughout the manuscript, filled in the boxplot, enlarged the axis, and completed the figure caption as well as defined the mature trees (M) more clearly in our methods (line 180) and figure caption.

Figure 7: incorrect y-axis description. Please correct

Response: Corrected the y-axis range from 0 to 100%

LL419-420: is that not a contradiction of what the MixSiar model shows in fig. 7? What does that mean?

Response: We provide a more detailed description of the change in depth to water uptake in October given the results of the MixSIAR model and isotopic composition of soil water at different depths (lines 485-487).

DISCUSSION

I think the discussion is too speculative and will be automatically improve once you can either provide flux data or direct the findings toward a different research question.

Response: We did a detailed revision of our discussion question to better address our revised hypotheses, decrease speculation on the root function of lodgepole pines, and build on our flux data findings.

LL 429- 430: how do you then explain the differences in soil water signatures shown in figure 4C in 07 and 09, where the blue (T2 treatment heavy thinning) boxes show a clear enrichment at the 5cm depth?

Response: We revised this sentence with a more subtle explanation of mixing between new and old precipitation events in the soils around 35 cm in agreement with the measured continuous soil moisture measurements (lines 498-501)

LL442-443 because this is the biggest seasonal water influx?? Relate to overall precip and summer precip?

Response: We changed this sentence to clarify that winter snow melt is the largest annual contributor to soil moisture (lines 515-516).

CONCLUSION:

This is a bit of a stretch and largely based on speculation about the fluxes in the discussion.

Response: We revised our entire conclusion section to decrease speculation, connect the climate, continuous soil moisture data, and isotopic composition of soil and tree water samples to connect our findings of seasonal changes in depth to water uptake to the larger concept of how changes in annual precipitation and lodgepole pine stand density influence the proportion and timing of depth to water uptake strategies.

References:

Allen, Scott T., and James W. Kirchner. 2022. 'Potential Effects of Cryogenic Extraction Biases on Plant Water Source Partitioning Inferred from Xylem-Water Isotope Ratios'. *Hydrological Processes* 36 (2): e14483. https://doi.org/10.1002/hyp.14483.

Response: This was a very useful citation to support our view that accounting for bias in isotopic extraction using cryogenic vacuum distillation would not affect our general conclusions.

---

## Author Response (AR2)

**Responses to comments from the Editor and reviewers**

\*\*\*\*\*\*\*\*\*\*\*\*\*\*\*\*\*\*\*\*\*\*\*\*\*\*\*\*\*\*\*\*\*\*\*\*\*\*\*\*\*\*\*\*\*\*\*\*\*\*\*\*\*\*\*\*\*\*\*\*\*\*\*\*\*\*\*\*\*\*\*\*\*\*\*\*\*\*\*\*\*\*\*

**Editor**

Comments:

Your manuscript received two substantially positive reviews, although corrections suggested by the referees may further improve its quality, please address them all.

Response:

Thank you for your positive feedback on our revision. Below you will see our complete responses to the comments and suggestions from two reviewers to further improve our manuscript during our second round of revisions.

\*\*\*\*\*\*\*\*\*\*\*\*\*\*\*\*\*\*\*\*\*\*\*\*\*\*\*\*\*\*\*\*\*\*\*\*\*\*\*\*\*\*\*\*\*\*\*\*\*\*\*\*\*\*\*\*\*\*\*\*\*\*\*\*\*\*\*\*\*\*\*\*\*\*\*\*\*\*\*\*\*\*

**Reviewer #1 Report #2**

Comments:

For final publications, the manuscript should be accepted as is. Were a revised manuscript to be sent for another round of reviews I would be willing to review the revised manuscript.

Response:

Thank you for your time and dedication to the publication of this manuscript. Your efforts are greatly appreciated.

```
* * *
```

Reviewer #2, Report #1

Comments:

For final publications, the manuscript should be accepted subject to minor revisions. Were a revised manuscript to be sent for another round of reviews I would be willing to review the revised manuscript.

I have now reviewed the revised version of the manuscript of Ellis et al. and acknowledge the significant improvements made by the authors. However, I believe there are still some minor points that should be addressed before publication.

Response:

Thank you for your diligent review and willingness to continue working on the revision process. Below are our response to your comments and suggestions.

Line 183: What does n = 3333 reflect? Maybe not related to this study?

Response:

In line 183, n = 3333 referred to the total number of samples (branches, soil, streams, and groundwater) collected over our sampling period. In response to your comment, we decided that it did not effectively contribute to the methods section and was misplaced in that sentence. Rather, Table 1 reflected a better overview of our sampling scheme for the summer of 2021. Because of this, we removed "(n=3333)" from our revised manuscript submission.

Line 236-238: Please clarify the definition of d2h "biases" and explain why these d2h biases are similar to those reported by Allen & Kirchner (2022).

Response:

We rephrased this sentence to address systematic biases as described by Allen & Kirchner that may occur during cryogenic vacuum distillation and broke up our leading sentence to provide more clarity on when bias occurs (see expert from lines 234-236 below).

> As reviewed by Allen & Kirchner (2022), the cryogenic vacuum distillation of water from plant tissues and soils can cause systematic biases in the measurements of $\delta^2H$. The degree of extraction bias varies depending on species and soil type (Allen & Kirchner, 2022).

Additionally, we added to the sentence connecting the findings of extraction bias by Allen & Kirchner (2022) to our results explaining the similar extraction methods and how any potential bias was smaller than the mean values of any potential sources (Lines 239-242).

Line 489-491: Please rephrase to clarify that this statement is based on isotope analysis.

Response:

Thanks. Following your suggestion, we now added "the results of our isotopic analysis" in Line 489-492 to clarify how we determined that there was more evaporative enrichment in the shallow soils of the heavily thinned stands and specified that this results was present when compared to the moderately thinned and control stands.

Line 495: Change "enrichment of d18O" to "enrichment in 18O."

Response:

Thank you for catching this. The correction has been made.

Line 461-466: The logic in this sentence is not entirely clear. I believe the authors intend to explain that Scenario 6 is the most realistic and statistically sound scenario. However, the text seems to "jump" between Scenarios 1, 2, and 6, as well as between Scenarios 4 and 6, without clear transitions. Additionally, Scenarios 3 and 5 are not mentioned, leaving gaps in the explanation. Could you please clarify the reasoning and ensure all relevant scenarios are addressed in a logical sequence?

Response:

Thank you for the comment. The objective of this paragraph is to describe the Gelman-Rubin diagnostic results of each of the scenarios and elaborate on why scenarios 4 and 6 were judged to be the most representative of different potential sources. In this revision (please see Lines 456-466), we broke up and reorganized this section to provide further clarity. Scenarios 1, 2, and 6 had the smallest Gelman-Rubin diagnostic results, indicating that they were the closest to converging. Additionally, we reiterated the variables considered for scenarios 4 and 6 to support why we identified these two scenarios as ones to rerun with a longer runtime.

Line 618-621: The current statement is not entirely clear. If thinning (or a decrease in stem density) increases soil water use below 35 cm during prolonged drought periods, then soil water below 35 cm becomes more relevant as the occurrence and duration of drought spells increase in the future (assuming this holds true for UPC). Therefore, lodgepole pine dependency on deep soil or groundwater would increase, not decrease as currently stated. Is this correct?

Response:

Lines 625-627: Great catch, thank you. The final sentence of that concluding statement has been revised to more clearly reflect the results of our study, which are that lodgepole pines are less dependent on summer precipitation events and more so on deep soil water and groundwater fed by winter snowpack/ spring snow melt rather than summer precipitation events under prolonged drought conditions.

Figures/Tables in general: Please explain the meaning of the lines in point plots as well as the boxes, whiskers, and symbols in boxplots. Additionally, provide more details on the location, species, and other relevant context. Ensure that all abbreviations are clearly defined. In general, each figure and table should be understandable on its own, without requiring the reader to refer to the entire manuscript.

Response:

We expanded the figure captions for figures 5, 6, 7, 8, and 9 to further explain the significance of boxplots and points in each figure as well as some of the results when appropriate to make the figures more comprehensive to individuals who have not read the entire manuscript.

Figure 9: The authors also collected isotope data from mature trees, but the rationale for including these data is not clearly expressed in the manuscript. I assume the mature trees are intended as a control to clarify the representativeness of the young tree water uptake patterns, correct? Additionally, I suggest adding the statistical results mentioned in Lines 453 and 455 to Figure 9 to make these key findings more explicit. Currently, these important results are somewhat "hidden" in the text, and highlighting them in the figure would enhance their visibility and impact.

Response:

Part of our expansion of the figure 9 caption is in response to the previously addressed comment, but we also included an explanation as to why the mature stand is only mentioned here and one of our key findings on changes in the isotopic signature of each of the stands between each sampling period. We also addressed some of the statistical findings and shift to a higher concentration of heavier isotopes in October when the values are the most positive in hopes of highlighting these results.

Figure 10: This is one of the most important figures in the paper. Could the authors please add more details to ensure the figure can be fully understood without needing to refer to the main

text? In particular, providing detailed information about Scenario 6 is crucial. Although the d2h biases are now mentioned in the Methods and Results sections, it remains unclear how these biases are considered in the MixSIAR model results. Please clarify how this bias is accounted for and its impact on the interpretation of the data.

Response:

We expanded our figure caption for this to include what sources and signatures were considered (in addition to reiterating it in the text) as well as the runtime parameters that were set in the MixSIAR model. We also described the more nuanced difference in the timing of changing proportions of depth to water uptake (more explicitly that the thinning treatments were able to maintain a larger proportion of shallow soil water uptake whereas the control stand shifted sources early in the study period).

The discussion of d2H bias was added to the main text (lines 498-501) to more explicitly state that it was not considered in our models.

Table S1: This table is not cited in the main text, or I may have missed the reference.

Response:

This supplemental table was not referenced in our previous submission but is now referenced in section 2.2. Climate and soil moisture monitoring (see line 177).